# Correlation of patient symptoms with SARS-CoV-2 Omicron variant viral loads in nasopharyngeal and saliva samples and their influence on the performance of rapid antigen testing

Kenichiro Shiraishi,[1] Yong Chong,[2] Takeyuki Goto,[1] Toshiyuki Ishimaru,[3] Nobuyuki Shimono,[4] Hideyuki Ikematsu,[5] Koichi Akashi[1]

**ABSTRACT** Evaluating SARS-CoV-2 viral loads in nasopharyngeal (NP) and saliva samples, factors affecting viral loads, and the performance of rapid antigen testing (RAT) have not been comprehensively conducted during SARS-CoV-2 Omicron epidemic. This prospective study included outpatients enrolled during Omicron variant period in Japan. Paired NP swab and saliva samples were collected to measure viral loads by reverse transcription-quantitative polymerase chain reaction (RT-qPCR). The correlation between viral loads and clinical symptoms was examined. The performance of an immunochromatography-based RAT kit was also assessed. A total of 153 patients tested within 3 days of symptom onset were included. The mean viral load was 5.60 $\log_{10}$ copies/test and 3.65 $\log_{10}$ copies/test in NP and saliva samples, respectively, resulting in a significant difference ($P < 0.0001$). Fever over 37°C (axillary temperature) and total number of symptoms other than fever were identified as independent factors positively correlated with the viral loads in both NP and saliva samples. RAT sensitivity using NP and saliva samples was 92% and 68%, respectively, using positive RT-qPCR results as the reference. The sensitivity of RAT using NP and saliva samples was significantly higher in patients with fever ≥37°C and/or at least one symptom than in those with fever <37°C and/or no symptoms (97% vs 83% in NP swabs; 80% vs 50% in saliva). Distinct symptoms, including fever ≥37°C, may reflect high Omicron variant viral loads. Rapid antigen testing, not only using nasopharyngeal swabs but also using saliva, would be useful for COVID-19 diagnosis as point-of-care testing, particularly for symptomatic patients.

**IMPORTANCE** We examined nasopharyngeal and salivary viral loads using samples collected from outpatients with SARS-CoV-2 infection during the Omicron epidemic in Japan and explored the outpatient factors correlated with viral loads. In addition, we evaluated the performance of an authorized rapid antigen testing (RAT) kit using nasopharyngeal and saliva samples with RT-PCR testing as the reference. Intriguingly, a correlation between fever and other symptoms and SARS-CoV-2 viral loads in nasopharyngeal and saliva samples was observed based on one COVID-19 outpatient visit. RAT sensitivity was influenced by viral loads. Nevertheless, nasopharyngeal RAT is considered useful for SARS-CoV-2 point-of-care diagnosis. In patients with distinct symptoms, including high-grade fever, salivary RAT could be a practical diagnostic tool because of the higher estimated viral loads. After the Omicron epidemic, outpatients with mild COVID-19 have become the main focus of diagnosis and treatment. Our study provides valuable information regarding the point-of-care diagnosis of these patients.

**KEYWORDS** SARS-CoV-2, Omicron variant, viral load, rapid antigen testing, patient symptom

Address correspondence to Yong Chong, jeong.yong.342@m.kyushu-u.ac.jp.

The authors declare no conflict of interest.

Since the end of 2021, the severe acute respiratory syndrome coronavirus 2 (SARS-CoV-2) Omicron variant lineages have been circulating worldwide (1). Point-of-care diagnosis and oral medication for outpatients with mild (Coronavirus disease 2019) (COVID-19) are becoming increasingly practical. Clinical symptoms, potentially associated with SARS-CoV-2 viral loads, could serve as a basis for outpatient diagnosis and treatment. The correlation between viral loads and severity in COVID-19 patients has been extensively investigated and has been inconclusive (2); however, reports on factors such as symptoms, which affect viral loads at symptom onset, have been limited, and little has been reported during the Omicron variant period.

Rapid antigen testing (RAT) using an immunochromatographic technique is now widely recognized and has been useful for the point-of-care diagnosis of COVID-19 in current clinical settings. The World Health Organization (WHO) and the Food and Drug Administration (FDA) recommend that RAT kits perform with at least 80% sensitivity and have approved several kits (3). Before the emergence of the Omicron variants, the sensitivity of clinically used RAT kits with NP swab samples was approximately 70% (4, 5). After the Omicron epidemic, changes in symptomatic characteristics and a reduction in severity compared with earlier strains have been documented (6, 7). These alterations suggest that the virological and clinical features of COVID-19 during the Omicron period may differ from those of the pre-Omicron era. The sensitivity of RAT kits using NP swabs during the Omicron period has shown a slight decrease, although results still seem inconclusive (3, 8). There are several advantages to using saliva for sample collection rather than NP swabs. NP swabs need to be collected by trained healthcare personnel wearing suitable protective equipment to prevent secondary infections. The risk of infection may be higher, particularly in patients with symptoms. In contrast, collecting saliva samples could contribute to reducing the burden and risk to healthcare personnel because self-collection is possible. The performance of RAT using saliva was examined during the pre-Omicron period; however, only a few reports are available (4). Furthermore, previous studies have shown performance discrepancies among their RAT kits using saliva (9–11).

The ALSONIC COVID-19 Ag immunochromatographic assay kit (Alfresa Pharma Corporation, Osaka, Japan) for NP swab samples was approved in Japan and released on March 18, 2021. It has been clinically used for point-of-care diagnosis of COVID-19 across Japan. This study formed part of a clinical trial aiming to gain additional approval for using the kit with saliva samples. We examined NP and salivary viral loads using samples collected from outpatients with SARS-CoV-2 infection during the Omicron epidemic in Japan and explored the outpatient factors (background and symptoms) correlated with viral loads. In addition, we evaluated the performance of the antigen test kit using NP and saliva samples with polymerase chain reaction (PCR) testing as the reference test. This comprehensive assessment was aimed to obtain useful information on current COVID-19 point-of-care settings.

## MATERIALS AND METHODS

### Participants

This study was conducted as a prospective clinical evaluation of the diagnostic accuracy of a newly developed rapid antigen test using saliva samples. Outpatients with fever and/or other symptoms were enrolled at participating clinics and hospitals in Japan from February 2022 to March 2023 during the Omicron epidemic period. Informed consent was obtained from the patient or from the parent if the patient was under 19 years of age.

### Study procedures

NP swab and saliva samples were collected for reverse transcription-quantitative PCR (RT-qPCR) and RAT. Briefly, a healthcare worker inserted a swab into the patient's nasal

cavity, stopped it upon encountering resistance, maintained the position for 10 seconds to allow absorption of nasal fluid, and then slowly removed the swab while rotating it. Two swabs were obtained from each patient and suspended in the specimen transport medium for RT-qPCR and in the extraction solution for RAT. The patients were instructed to collect 2.5 mL of saliva, which was naturally pooled in the mouth, in a container. Patients avoided collecting saliva immediately after eating, drinking, brushing their teeth, or gargling, ensuring a minimum interval of 30 min between these activities. Two swabs were permeated with saliva and suspended in the specimen transport medium for RT-qPCR and in the extraction solution for RAT. Specimens for RT-qPCR were transported under refrigeration, stored at −80℃ until analysis, and tested in a central laboratory.

The number of days from symptom onset to sample collection was recorded. The day of symptom onset was defined as day 1 after symptom onset. Information was collected on the participant's background, including their history of COVID-19 vaccination. Axillary body temperature, routinely used in Japan, was measured at the time of sample collection. A temperature of ≥37℃ was defined as symptomatic fever. The presence or absence of other symptoms, including cough, sore throat, nasal discharge, fatigue, and dysgeusia, was confirmed by questioning by healthcare workers. The remaining four symptoms excluding dysgeusia were used to calculate total symptom scores. Total symptom scores were calculated with the presence of each symptom as one point.

## RT-qPCR

RT-qPCR detection of SARS-CoV-2 was conducted using NP swabs and saliva. RNA extraction was performed using a sample volume of 140 µL with the QIAamp Viral RNA Mini Kit (QIAGEN, Hilden, Germany) according to the manufacturer's protocol and the COVID-19 Pathogen Detection Manual, ver.2.9.1(12). RT-qPCR was performed using the QuantiTect Virus + ROX Vial Kit (QIAGEN). The detection of SARS-CoV-2 genes was performed using the TaqMan probe method. The primers and probe targeting the nucleocapsid protein, as listed in the COVID-19 Pathogen Detection Manual, ver.2.9.1 (12), were as follows: forward primer (5′-AAATTTTGGGGACCAGGAAC-3′); reverse primer (5′-TGGCAGCTGTGTAGGTCAAC-3′); and probe (5′-FAM-ATGTCGCGCATTGGCATGG A-BHQ-3′). The assessment of positive and negative results was performed according to the COVID-19 Pathogen Detection Manual, ver.2.9.1, and Guidelines for the operation of assay for the novel coronavirus (SARS-CoV-2), third edition (12, 13). The viral load of each sample was calculated based on a calibration curve established using a 10-fold dilution series of the positive control RNA (nucleocapsid region) containing $1.0 \times 10^6$ copies/µL with the Positive Control RNA Mix (2019-nCoV) (Takara Bio Inc., Kusatsu, Japan) (14).

## Identification of SARS-CoV-2 variants

Gene sequences of the Wuhan and variant SARS-CoV-2 strains were detected using the TaqMan probe-based RT-qPCR with One Step PrimeScript III RT-qPCR Mix (Takara). The primers and probes were used according to the Primer/Probe Set 2019-nCoV (Takara). Variant strains were identified based on amino acid (AA) substitutions of AAs 339, 452, 484, and 501.

## Rapid antigen testing

We used the ALSONIC COVID-19 Ag immunochromatographic assay kit (Alfresa Pharma). The swab containing the specimen was immersed in the extraction solution and agitated multiple times by inversion. Three drops of the extract (approximately 60 µL) were applied to a test cassette; the test results were determined by the presence or absence of a band on the kit visually observed after 5 min at room temperature. The determination of band positivity using the kit was performed by physicians at each participating institution, who were blinded to the PCR test results.

## Statistical analysis

The sensitivity, specificity, positive predictive value (PPV), and negative predictive value (NPV) were calculated. The viral load value was log-transformed using a base value of 10. Categorical variables were analyzed using the Chi-squared test. Student's *t*-test was used to compare continuous variables between the two groups, whereas ANOVA was applied among three or more groups. A paired *t*-test was specifically utilized for comparing paired data between two groups. Spearman's rank-correlation test was used to assess correlations between two data sets. Multivariate linear regression models were established using a stepwise selection procedure. The level of significance was set at $P < 0.05$, 2-sided. All analyses were performed using the JMP Pro, version 7.0 (SAS Institute, Cary, NC, USA).

## RESULTS

A total of 193 outpatients were enrolled during the study period, and 39 patients tested ≥4 days (day 4) after the onset of symptoms were excluded because of the small number of SARS-CoV-2-positive cases (<10 cases per day). Of the remaining patients, one was excluded due to a lack of paired collection of NP swab and saliva samples. In this study, SARS-CoV-2 infection was considered present if either NP or salivary RT-qPCR result was positive. A total of 153 patients were included in the analysis, of whom 117 were SARS-CoV-2-positive and 36 were negative. Of the 117 SARS-CoV-2-positive samples, 105 (89.7%) were identified as Omicron variants (6 Omicron BA.1/BA.2 variants, 99 Omicron BA.4/BA.5 variants, and 12 unidentifiable strains). The mean age of SARS-CoV-2-positive patients was 42.4 years, 45.3% were male, and there were no patients with cancer or immunosuppressed status.

The viral loads of paired NP and saliva samples of SARS-CoV-2-positive patients are depicted in Fig. 1. Overall, the viral load in saliva was significantly lower than that in NP swabs ($P < 0.0001$). The mean viral load in NP swab and saliva samples was 5.60 $\log_{10}$ copies/test (95% CI, 5.30–5.91) and 3.65 $\log_{10}$ copies/test (95% CI, 3.39–3.91), respectively. The mean difference in viral loads between NP swab and saliva samples was 1.95 $\log_{10}$ copies/test (95% CI, 1.66–2.25). In each analysis based on the number of days from symptom onset, a similar tendency was observed in the comparison of viral loads between NP swabs and saliva. The correlation between viral loads in NP swab and saliva samples is shown in Fig. S1. The viral load in saliva samples was significantly correlated with that in NP swab samples ($P < 0.0001$).

To examine factors related to SARS-CoV-2 viral loads, we analyzed the characteristics of NP and salivary viral loads according to the patient's background and symptoms (Table 1). In the univariate analysis, fever ($P = 0.049$), cough ($P = 0.031$), and symptom scores ($P = 0.032$) were significantly correlated with the NP viral load, and sex ($P = 0.012$), fever ($P = 0.001$), fatigue ($P = 0.004$), and symptom scores ($P = 0.002$) showed a significant correlation with the salivary viral load. In the multivariate analysis, fever (standardized regression coefficient: $\beta = 0.223$ [95% CI, 0.069–0.806]; $P = 0.02$) and symptom scores ($\beta = 0.194$ [95% CI, 0.02–0.616]; $P = 0.037$) were extracted as being independently correlated with the NP viral load. A similar analysis for the salivary viral load extracted fever ($\beta = 0.272$ [95% CI, 0.163–0.751]; $P = 0.003$) and symptom scores ($\beta = 0.277$ [95% CI, 0.151–0.627]; $P = 0.002$) as independently significant.

All 153 paired samples collected were tested using both RT-qPCR testing and RAT. Salivary RT-qPCR showed 95.7% sensitivity and 97.3% specificity, using the paired NP RT-qPCR result as the reference. The diagnostic performances of NP and salivary RAT, based on the SARS-CoV-2 infection status determined by RT-qPCR, are shown in Table 2. The sensitivity of NP and salivary RAT was 91.5% and 68.4%, respectively, with a specificity of 100% for both sample types. The sensitivity of NP and salivary RAT according to the viral load in each sample is shown in Table 3. Samples with viral loads ≥ $1.0 \times 10^3$ copies/test showed sensitivities ≥ 80% in both NP and salivary RAT. In contrast, the sensitivity was considerably lower in samples with viral loads < $1.0 \times 10^3$ copies/test, regardless of the sample type. Table 4 shows NP and salivary viral loads and

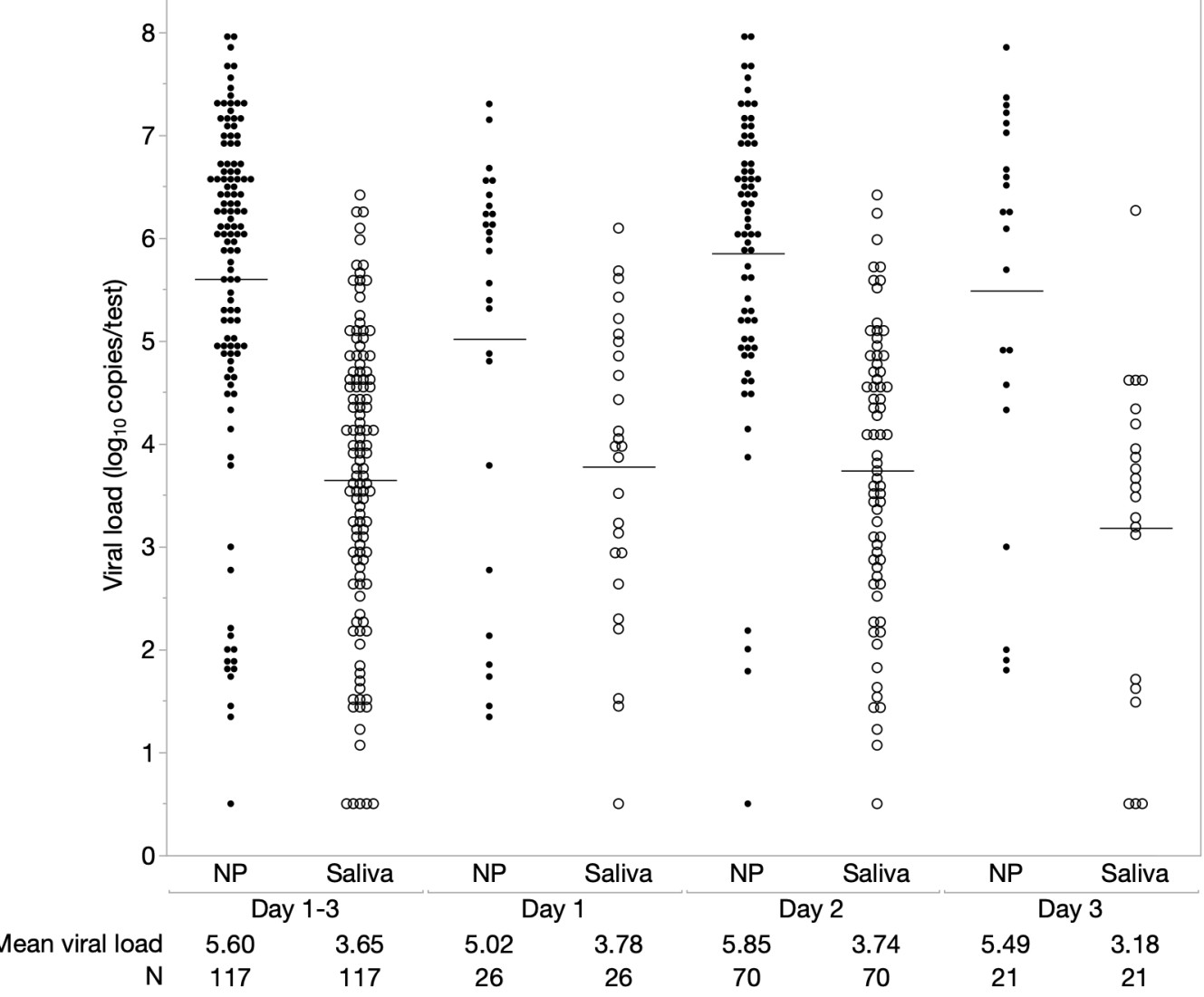

| | NP | Saliva | NP | Saliva | NP | Saliva | NP | Saliva |
|---|---|---|---|---|---|---|---|---|
| | \multicolumn Day 1-3 | | Day 1 | | Day 2 | | Day 3 | |
| Mean viral load | 5.60 | 3.65 | 5.02 | 3.78 | 5.85 | 3.74 | 5.49 | 3.18 |
| N | 117 | 117 | 26 | 26 | 70 | 70 | 21 | 21 |

FIG 1 Viral loads of paired nasopharyngeal and saliva samples in SARS-CoV-2-positive patients based on the number of days from symptom onset. The black and white circles represent data obtained from nasopharyngeal and saliva samples, respectively. The horizontal solid bars represent the mean values. Paired *t*-test was used to compare paired nasopharyngeal and salivary data. NP, nasopharyngeal.

RAT sensitivity according to the presence or absence of symptoms. NP and salivary viral loads tended to be higher in patients with fever ≥37°C and/or symptom scores ≥ 1 than in those with <37°C and/or no symptom scores, with a significant difference observed in saliva. The sensitivity of RAT using NP and saliva samples was significantly higher in patients with fever ≥37°C and/or total symptom scores ≥ 1 than in those with fever <37°C and/or no symptom scores (97% vs 83% in NP swabs, $P = 0.006$; 80% vs. 50% in saliva, $P < 0.001$).

## DISCUSSION

In this study, we measured and compared SARS-CoV-2 NP and salivary viral loads to use these data as a reference and to leverage the benefits of saliva-based testing. The viral load in saliva was significantly lower than that in NP swabs, and the difference in viral loads between NP and saliva samples was approximately two $\log_{10}$ copies/test. Previous studies conducted during the Omicron variant epidemic have reported lower viral loads in saliva than in NP swabs, with a difference of approximately 10× to 100× (9, 11, 15–17). In previous studies, a similar tendency between NP and salivary viral loads was reported

**TABLE 1** Influence of patient demographics and symptoms on SARS-CoV-2 viral loads in nasopharyngeal and saliva samples[d]

| Characteristic | n (%) | NP viral load ($\log_{10}$ copies/test) | | | Salivary viral load ($\log_{10}$ copies/test) | | |
|---|---|---|---|---|---|---|---|
| | | Mean | 95% CI | *P*-value | Mean | 95% CI | *P*-value |
| Total[a] | 117 | 5.60 | 5.30–5.91 | | 3.65 | 3.39–3.91 | |
| Sex | | | | | | | |
| Female | 64 (54.7) | 5.59 | 5.17–6.00 | | 3.35 | 3.00–3.69 | |
| Male | 53 (45.3) | 5.62 | 5.16–6.07 | 0.916 | 4.01 | 3.63–4.39 | 0.0118 |
| Age (years) | | | | | | | |
| ≤29 | 18 (15.4) | 5.4 | 4.62–6.18 | | 3.67 | 2.99–4.34 | |
| 30–59 | 85 (72.6) | 5.59 | 5.23–5.95 | | 3.67 | 3.36–3.98 | |
| ≥60 | 14 (12.0) | 5.92 | 5.03–6.81 | 0.684 | 3.51 | 2.75–4.28 | 0.933 |
| Fever[b] | | | | | | | |
| <37.0°C | 45 (38.5) | 5.24 | 4.75–5.72 | | 3.10 | 2.70–3.50 | |
| 37.0–37.9°C | 33 (28.2) | 5.50 | 4.93–6.06 | | 3.74 | 3.27–4.21 | |
| ≥38.0°C | 39 (33.3) | 6.11 | 5.59–6.63 | 0.0492 | 4.20 | 3.77–4.63 | 0.0014 |
| Cough | | | | | | | |
| Negative | 50 (42.7) | 5.22 | 4.76–5.68 | | 3.37 | 2.98–3.77 | |
| Positive | 67 (57.3) | 5.89 | 5.49–6.28 | 0.0309 | 3.85 | 3.51–4.19 | 0.0724 |
| Sore throat | | | | | | | |
| Negative | 31 (26.5) | 5.36 | 4.77–5.95 | | 3.33 | 2.83–3.84 | |
| Positive | 86 (73.5) | 5.69 | 5.33–6.04 | 0.349 | 3.76 | 3.46–4.07 | 0.151 |
| Nasal discharge | | | | | | | |
| Negative | 76 (65.0) | 5.58 | 5.20–5.96 | | 3.48 | 3.16–3.80 | |
| Positive | 41 (35.0) | 5.64 | 5.12–6.16 | 0.853 | 3.95 | 3.52–4.39 | 0.0888 |
| Fatigue | | | | | | | |
| Negative | 44(37.6) | 5.23 | 4.74–5.73 | | 3.16 | 2.75–3.58 | |
| Positive | 73(62.4) | 5.82 | 5.44–6.21 | 0.0842 | 3.94 | 3.62–4.26 | 0.0039 |
| Dysgeusia | | | | | | | |
| Negative | 115 (98.3) | 5.60 | 5.30–5.91 | | 3.66 | 3.40–3.93 | |
| Positive | 2 (1.7) | 5.46 | 3.12–7.81 | 0.907 | 2.79 | 0.79–4.79 | 0.768 |
| Total symptom scores[c] | | | | | | | |
| 0 | 4 (3.4) | 3.32 | 1.71–4.92 | | 1.64 | 0.31–2.98 | |
| 1 | 20 (17.1) | 5.56 | 4.85–6.28 | | 3.17 | 2.57–3.77 | |
| 2 | 48 (41.0) | 5.47 | 5.01–5.93 | | 3.63 | 3.25–4.02 | |
| 3 | 29 (24.8) | 5.85 | 5.26–6.45 | | 3.82 | 3.33–4.32 | |
| 4 | 16 (13.7) | 6.17 | 5.36–6.97 | 0.0319 | 4.48 | 3.81–5.15 | 0.0021 |
| Vaccination (number of doses) | | | | | | | |
| 0 | 10 (8.5) | 5.66 | 4.61–6.71 | | 3.81 | 2.91–4.71 | |
| one or 2 | 29 (24.8) | 5.82 | 5.20–6.43 | | 3.64 | 3.11–4.17 | |
| ≥3 | 78 (66.7) | 5.51 | 5.14–5.89 | 0.704 | 3.63 | 3.31–3.95 | 0.931 |

[a]Patients tested within 3 days of symptom onset were included.
[b]Axillary body temperature was measured.
[c]Four symptoms excluding fever and dysgeusia were used to calculate total symptom scores.
[d]NP, nasopharyngeal; CI, confidence interval.

using samples collected during wild-type and Alpha/Delta variant epidemics (15–17). A lower SARS-CoV-2 viral load in saliva than in NP swabs is likely to be observed, regardless of the type of epidemic strains. Several reasons for this quantitative relationship between NP swabs and saliva include the viscosity of saliva, which can affect the sensitivity of reagents, the presence of reaction inhibitors in saliva, and the presence of defective and non-viable viruses in NP swabs (11, 15). However, the definite reasons require further investigation. The sensitivity of salivary RT-qPCR performed in our study was extremely high (95.7%) and was shown to be useful for SARS-CoV-2 detection during the Omicron variant period. The performance of RT-qPCR was little affected by the difference in viral loads between NP swab and saliva samples.

**TABLE 2** Performance of rapid antigen testing for SARS-CoV-2 diagnosis[a,b,c]

|  | Sensitivity (%, n) | Specificity (%, n) | PPV (%, n) | NPV (%, n) |
|---|---|---|---|---|
| NP RAT | 91.5 (107/117) | 100.0 (36/36) | 100.0 (107/107) | 78.3 (36/46) |
| Salivary RAT | 68.4 (80/117) | 100.0 (36/36) | 100.0 (80/80) | 49.3 (36/73) |

[a]Patients tested within 3 days of symptom onset were included.
[b]RAT, rapid antigen testing; NP, nasopharyngeal.
[c]PPV, positive predictive value; NPV, negative predictive value.

In our prospective study, fever and the number of symptoms were identified as independent factors positively correlated with SARS-CoV-2 viral loads in both NP and saliva samples. Studies on the correlation between viral loads and symptoms in COVID-19 patients are limited, in contrast to reports on the effect of viral loads on disease severity (2). Previous retrospective studies conducted before the Omicron variant epidemic suggested a possible correlation between the presence or absence of fever or other symptoms and NP viral loads using univariate analysis (18, 19). A prospective study performed during the Omicron variant epidemic suggested a significant correlation between NP viral loads and the presence or absence of fever using univariate analysis (9). Notably, our finding of a correlation between the grade of fever and the number of symptoms with viral loads was based on data from one outpatient visit at symptom onset. Thus, information regarding the grade of fever or other symptoms during outpatient visits could be useful for estimating the amount of SARS-CoV-2 in NP and saliva samples.

A recent meta-analysis evaluating the performance of RAT using NP swab samples during the Omicron period reported a sensitivity of 67.1% and a specificity of 100% (3). In addition, WHO/FDA-approved, widely utilized Abbott kits have demonstrated a sensitivity of approximately 70% with NP samples (3). Based on these reports, the RAT kit used in our study has shown excellent performance when using NP samples. The performance assessments of salivary RAT in combination with NP RAT have been limited, with a large variability of sensitivity in NP vs saliva samples reported in different studies (66% vs 2%, 68% vs 41%, and 93% vs 74%) (9–11). The performance of RAT used in our study (92% vs. 68%) was similar to that reported by Kodana et al. (11). Our findings regarding the NP RAT with high sensitivity and specificity suggest that RAT using NP samples is useful for point-of-care testing, as a simple and rapid alternative to RT-qPCR. In our study, >80% sensitivity with salivary RAT was obtained for samples with SARS-CoV-2 viral loads $\geq 1.0 \times 10^3$ copies/test; however, its sensitivity was considerably lower at viral loads $< 1.0 \times 10^3$ copies/test. The sensitivity of salivary RAT was greatly affected in patients with a salivary viral load in the lowest tertile ($< 1.0 \times 10^3$ copies/test). A similar finding of higher RAT sensitivity in samples with higher viral loads has been reported previously (20, 21). In our study, NP and salivary RAT sensitivities were significantly higher in patients with fever ≥37°C and/or symptoms than in those with low-grade fever and/or few symptoms (97% vs 83% in NP swabs; 80% vs 50% in saliva). The

**TABLE 3** Sensitivity of nasopharyngeal and salivary rapid antigen testing according to viral loads[a,b]

| Viral load (copies/test) | NP RAT | | Salivary RAT | |
|---|---|---|---|---|
|  | Sensitivity (%) | Positive (n) / Total (n) | Sensitivity (%) | Positive (n) / Total (n) |
| $10^7$–$10^8$ | 100.0 | 21/21 | N/A | N/A |
| $10^6$–$10^7$ | 100.0 | 43/43 | 100.0 | 4/4 |
| $10^5$–$10^6$ | 100.0 | 20/20 | 100.0 | 15/15 |
| $10^4$–$10^5$ | 100.0 | 17/17 | 100.0 | 33/33 |
| $10^3$–$10^4$ | 100.0 | 2/2 | 83.3 | 25/30 |
| $10^2$–$10^3$ | 40.0 | 2/5 | 16.7 | 3/18 |
| $10^1$–$10^2$ | 25.0 | 2/8 | 0.0 | 0/12 |
| 0–10 | 0.0 | 0/1 | 0.0 | 0/5 |
| Total | 91.5 | 107/117 | 68.4 | 80/117 |

[a]Patients tested within 3 days of symptom onset were included.
[b]RAT, rapid antigen testing; NP, nasopharyngeal; N/A, not available.

**TABLE 4** Nasopharyngeal and salivary viral loads and rapid antigen testing sensitivity according to the presence or absence of patient symptoms[a,c]

| | Mean viral load ($\log_{10}$ copies/test) | | | | Sensitivity of RAT (%, n) | | | |
|---|---|---|---|---|---|---|---|---|
| | NP | *P*-value | Saliva | *P*-value | NP | *P*-value | Saliva | *P*-value |
| Fever <37°C and/or no symptom scores[b] | 5.23 | | 3.08 | | 82.6 (38/46) | | 50.0 (23/46) | |
| Fever ≥37°C and/or total symptom scores ≥ 1 | 5.84 | 0.075 | 4.02 | 0.0004 | 97.2 (69/71) | 0.0059 | 80.3 (57/71) | 0.0006 |

[a]Patients tested within 3 days of symptom onset were included.
[b]Axillary body temperature was measured. Four symptoms excluding fever and dysgeusia were used to calculate symptom scores.
[c]RAT, rapid antigen testing; NP, nasopharyngeal.

sensitivity of RAT using NP swab samples has been reported to be higher in symptomatic than in asymptomatic patients (20, 22). Currently, following the Omicron epidemic, outpatient services for COVID-19 have become more common owing to the reduction in its severity. RAT limited to patients with distinct symptoms may be a practical option for point-of-care testing in clinical settings. The risk of secondary infections during sample collection is potentially higher in symptomatic patients than in asymptomatic patients. The diagnosis of COVID-19 using salivary RAT may be useful, particularly in patients with symptoms.

A limitation of our study was its small sample size. Based on the sample size, the conclusions and generalizability of this study are limited. Patient symptom information was collected only at the time of the initial visit and not recorded sequentially. Additionally, data on the grade of symptoms other than fever were not obtained. We did not examine the presence or absence of backgrounds that might be associated with patient symptoms. Consequently, adjusting for potential confounding factors that could influence the reported symptoms was challenging. The Omicron lineages that have mutated from the BA variants, during of which our study was conducted, have been circulating. Further studies on the relative effects of current omicron subvariants are warranted. Despite these limitations, the results of this study are clinically relevant. Intriguingly, a correlation between fever and other symptoms and SARS-CoV-2 viral loads in NP and saliva samples was observed based on one COVID-19 outpatient visit. RAT sensitivity tended to be influenced by viral loads. Nevertheless, RAT using NP samples is considered useful for SARS-CoV-2 point-of-care diagnosis. In patients with distinct symptoms, including high-grade fever, RAT using saliva samples appears to be a practical diagnostic tool because of the higher estimated viral loads. After the Omicron variant epidemic, outpatients with mild COVID-19 have become the main focus of diagnosis and treatment. Our study provides valuable information regarding the point-of-care diagnosis and treatment of these patients.

## ACKNOWLEDGMENTS

The authors thank the following doctors for participating in this study: Dr. Jun Matsumura, Dr. Shizuo Shindo, and Dr. Takuma Bando.

This study was funded by Alfresa Pharma Co., Ltd (Osaka, Japan).

K.S., Y.C., and T.G. performed data analysis and interpretation. K.S. and Y.C. drafted the manuscript, and Y.C., T.I., N.S., H.I., and K.A. revised the manuscript.

## AUTHOR AFFILIATIONS

[1]Medicine and Biosystemic Science, Kyushu University Graduate School of Medical Sciences (The First Department of Internal Medicine), Fukuoka, Japan
[2]Department of Clinical Immunology, Rheumatology, and Infectious Disease, Kyushu University Hospital, Fukuoka, Japan
[3]Department of Infectious diseases, Japanese Red Cross Fukuoka Hospital, Fukuoka, Japan
[4]Center for the Study of Global Infection, Kyushu University Hospital, Fukuoka, Japan
[5]Ricerca Clinica Co., Fukuoka, Japan

## AUTHOR ORCIDs

Yong Chong  http://orcid.org/0000-0002-8804-3181

## ETHICS APPROVAL

This study was approved by the institutional review board of Hara-Doi Hospital (Approval Date: January 18, 2022).

## ADDITIONAL FILES

The following material is available online.

### Supplemental Material

**Figure S1 (Spectrum00932-24-s0001.tiff).** Correlation between viral loads of paired nasopharyngeal and saliva samples in SARS-CoV-2-positive patients.

### Open Peer Review

**PEER REVIEW HISTORY (review-history.pdf).** An accounting of the reviewer comments and feedback.

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
