## [Reviewer comments · Microbiology Spectrum]

Microbiology Spectrum

Correlation of patient symptoms with SARS-CoV-2 Omicron variant viral loads in nasopharyngeal and saliva samples and their influence on the performance of rapid antigen testing

Kenichiro Shiraishi, Yong Chong, Takeyuki Goto, Toshiyuki Ishimaru, Nobuyuki Shimono, Hideyuki Ikematsu, and Koichi Akashi

Corresponding Author(s): Yong Chong, Kyushu Daigaku Byoin

Review Timeline:

Submission Date:	April 12, 2024
Editorial Decision:	June 20, 2024
Revision Received:	July 27, 2024
Editorial Decision:	August 19, 2024
Revision Received:	August 26, 2024
Accepted:	September 7, 2024

Editor: Sophia Georghiou

Reviewer(s): The reviewers have opted to remain anonymous.

Transaction Report:

DOI: <https://doi.org/10.1128/spectrum.00932-24>

Re: Spectrum00932-24 (Correlation of patient symptoms with SARS-CoV-2 Omicron variant viral loads in nasopharyngeal and saliva samples and their influence on the performance of rapid antigen testing)

Dear Dr. Yong Chong:

Thank you for the privilege of reviewing your work. Below you will find my comments, instructions from the Spectrum editorial office, and the reviewer comments.

Revision Guidelines

Sincerely,
Sophia Georghiou
Editor
Microbiology Spectrum

Reviewer #1 (Comments for the Author):

Major comments

1. Though not mentioned in the limitation, the information regarding vaccination status (times and the duration from the last shot) were not described in this manuscript. This is critical, as the presence or absence of vaccine history could affect both decreased viral load and mitigated symptoms. Clarifying the correlation of the vaccine status and other clinical information including viral load and symptoms would fundamentally have impact on the message of this study.
2. Though this is a multicenter study involved by several medical center, the details regarding sample transportation and the

location where the testing were performed (central testing or each facility?) are not specified in the Method section. In addition, more information about the detailed method for RT-PCR should be described, including nucleic acid extraction method, sample volume, and target region. These factors would affect the results of RT-PCR, especially quantitative results.

Minor comments

1. The context in the Introduction 56-60 is difficult to understand in current form. These sentences should be restructured for logical flow.
2. I cannot agree the description in the Introduction 63 "RAT for the detection of SARS-CoV-2 has received limited evaluation" based on the citation stated in 2020, since a number of studies have reported the evaluation of RAT to date.
3. Approval number of the IRB not specified in the text (different institution from the author's affiliation?).
4. The authors described the Identification of SARS-CoV-2 variants in the Method section, whose corresponding result were not found.
5. About RAT used in this study, whether the use of saliva specimen is approved or not should be mentioned. How the decision was made (single or double, open or blind) should be noted.
6. The statistical analysis applied for each analysis is not clear. About the analysis comparing NP and saliva viral load, paired test would fit better in this study.
7. Poor information in figure legends, including details about statistical analysis.

Reviewer #2 (Comments for the Author):

It was with great interest that I read the original manuscript entitled "Correlation of patient symptoms with SARS-CoV-2 Omicron variant viral loads in nasopharyngeal and saliva samples and their influence on the performance of rapid antigen testing" by Shiraishi et al. presented to Microbiology Spectrum.

This manuscript covers a clinical validation study of saliva-based testing for Omicron SCV-2 variants, using both traditional RT-qPCR testing as well as a rapid antigen test. The statistics and comparisons used in the study are appropriate. The authors use a small, but well characterized cohort in the study.

The introduction does appear dated and I suspect it may have been written, at least in part, several years ago. For example, statements such as (Line 61): "RAT using an immunochromatographic technique is expected to be useful for POC diagnosis of COVID-19 in clinical settings". I would argue, this statement should be in the past tense, and that RATs have been widely used and have shown value in at-home and clinical diagnosis of COVID-19. I would also argue that the statement (Line 64) that RATs using saliva have hardly been validated is incorrect and at this point has been heavily and widely investigated across multiple technologies with multiple reviews and meta-analyses on the topic. The authors should update the introduction using more recent data and references. Because the authors focus on Omicron, additional background information on clinical presentation, viral loads, validation studies on the Omicron variant should be included (there are vast data on this topic).

The only true novelty in this manuscript is the validation of rapid test for which little clinical data is available. Therefore, the authors should in their discussion, compare the performance of their rapid tests to other technologies and RATs that are more widely used and distributed - for example. does this RAT appear more, similar, or less sensitive to these other technologies? Additionally, a few additional lines in the introduction on this test should be included - was it / where is it approved, when was it released, where and how frequently is it used, etc.

Minor comments:

The use of axillary temperature and the 37 degree cut-off for denoting fever should be justified in the methods. In the abstract and tables, when denoting fever and non-fever temperature cut-offs it should be clarified that this is through axillary measurement.

Throughout the manuscript RT-PCR should be RT-qPCR or qRT-PCR - the RT here should denote reverse transcription not real-time as defined in the manuscript. Because of your chosen terminology some sentences are illogical like (Line 109): "Qualitative and quantitative RT-PCR detection" - as real-time and quantitative are terms that are used interchangeably. As an additional example, you use the RT-qPCR Mix in your methods (Reverse Transcription quantitative PCR), where RT denotes Reverse Transcription.

Line 119: What control materials were used for positive controls? Were controls used during the extraction step?

Line 130-135: Were test operators aware of qRT-PCR results when interpreting RATs or were results blinded? Did a single operator interpret all RAT test results?

Line 153-154: Where any other confounding metrics assessed that may be associated with certain symptomology (e.g. smoking with cough)?

Line 395: NP = Nasopharyngeal as defined in main text body.

The authors present a combined PCR / lateral flow assay for the simultaneous detection of three zoonotic pathogens. The authors provide a preliminary analytical validation of their assay using in vitro transcribed RNAs, spiked into biological samples collected from bat, pig, or camel. They also investigate possible interactions between the different tags used in their assay. Throughout the pandemic I have been involved in the validation and distribution of COVID-19 rapid test supplies to resource limited settings. I am having some difficulty understanding the utility of this assay in these settings. There are a few limitations that I believe need to be addressed in the manuscript when considering the deployment of this technology.

A major concern is the risk of contamination - this assay requires PCR amplification and subsequently detection of amplification using lateral flow. Because of this, there is a substantial risk of contamination when the user manipulates the PCR product to load onto the gel.

- The authors should acknowledge this limitation and make recommendations to address the issue (UNG system, compartmentalized workflows, and negative controls).

- How do they recommend this transfer step is accomplished in resource limited settings?

The authors should also highlight as additional limitations in the introduction that their proposed solution is a qualitative test and the impact of possible primer dimerization on specificity.

I believe the authors also need to improve their justification for the selected solution as compared to some of the other options available. They also need to discuss its limitations in addition to its strengths as compared to other approaches to the same problem. For example, why use this technique instead of qPCR? Some compact qPCR machines are available that are fairly affordable (~6000 USD), that would be easily multiplexed, avoid the risk of contamination because tubes would not need to be opened, would be quantitative, would not require a secondary detection step, would be able to run many samples at once, would be more sensitive, and the results wouldn't require individual user interpretation. The justification that 'qPCR machines are expensive' is simply not enough. Alternatively, isothermic LAMP or NDA technologies for amplification would require almost no laboratory equipment other than an incubator which could be accomplished with a small and low footprint battery-powered device eliminating the need for a PCR machine altogether. These could also be paired with a dye indicator or fluorescence for detection to remove the need for a secondary detection step, decreasing hands on time and minimizing the risk of contamination. The authors should weigh the pros and cons of their proposed solution against these others, not against gel electrophoresis.

I find it suspicious/concerning that one of their assays would have a limit of detection <10 copies, whereas the other two assays show a limit of detection >10,000 copies. This 1000-fold difference in sensitivity suggests possible issues with the quantification of the plasmids, or technical issues with two of the assays. The authors should address this in their discussion. Where these experiments performed in singlicate or replicate? Do the authors have access to digital PCR as a more robust method for quantifying their IVT products? Does this have to do with a higher than recommended annealing temperature to avoid primer dimerization? Please touch on this in the discussion.

For qualitative tests that require operator interpretation such as that developed here, it's important that the interpreter of the test result is unaware of the suspected outcome. Was there any attempt in this study to blind sample identity to prevent a user-bias?

Minor comments:

Importance: I'm surprised with the ongoing COVID-19 pandemic that the authors don't use this as justification for the need for detection of zoonotic pathogens in different reservoirs. This provides strong support for the surveillance for coronaviruses and other pathogens in resource-limited settings.

Material and methods: Please provide details on the RNA extraction process. How would this be accomplished in resource limited settings, and are there any health/safety risks (i.e. how can trizol be used safely in resource limited settings)?

Throughout the manuscript: 'Immuno-assay' should be one word as 'immunoassay'.

Figures: Many of these figures and tables could be moved to the supplement. Please cut down to the 4-6 most informative/impactful figures and tables and move the rest to the supplement.

Supplemental table 1: I think these are some of the more important experiments in the manuscript, showing the ability of the assay to specifically detect viral RNA in different tissues. However, I can't find any information in the manuscript about the concentration of RNA that was spiked (other than the mass of RNA) into each of the treatments - this information should be included in the table or table description to ease interpretation. The false positive and false negative rate should also be included in this table. Additional replicates at either high or low concentrations of spiked virus should be included if they were not already.

Responses to Reviewer 1's comments

Major comments

1. The reviewer indicated, "Though not mentioned in the limitation, the information regarding vaccination status (times and the duration from the last shot) were not described in this manuscript." We thank you for the suggestion. In the initial submission, we reported the viral load results by the presence or absence of vaccination in Table 1, which showed no significant differences. Based on the reviewer's suggestion, we have examined the correlation of the number of vaccine doses with viral loads. This additional categorization by the number of vaccine doses revealed no correlation with NP or salivary viral loads. Unfortunately, we could not analyze the correlation of the duration from the last dose with viral loads due to a lack of data on the last vaccination dates. We have revised the presentation of vaccination history in Table 1 from the initial categorization to one base on the number of vaccine doses.

2-1. The reviewer indicated, "Though this is a multicenter study involved by several medical center, the details regarding sample transportation and the location where the testing were performed (central testing or each facility?) are not specified in the Method section." Specimens for RT-PCR were transported under refrigeration, stored at -80°C until analysis, and tested in a central laboratory. According to the reviewer's suggestion, we have added the following statement to the Methods section: Page 6, lines 125 to 127: Specimens for RT-qPCR were transported under refrigeration, stored at -80°C until analysis, and tested in a central laboratory.

2-2. The reviewer indicated, "In addition, more information about the detailed method for RT-PCR should be described, including nucleic acid extraction method, sample volume, and target region. These factors would affect the results of RT-PCR, especially quantitative results." Based on the reviewer's recommendation, we have revised the statements concerning the nucleic acid extraction method and RT-PCR in the Methods section as follows:

Page 7, lines 139 to 142: RNA extraction was performed using a sample volume of $140\ \mu\text{L}$ with the QIAamp Viral RNA Mini Kit (QIAGEN, Hilden, Germany) according to the manufacturer's protocol and the COVID-19 Pathogen Detection Manual, ver.2.9.1(12).

Page 7, lines 144 to 149: The primers and probe targeting the nucleocapsid protein, as listed in the COVID-19 Pathogen Detection Manual, ver.2.9.1 (12), were as follows: forward primer (5'-AAATTTTGGGGACCAGGAAC-3'); reverse primer (5'-TGGCAGCTGTGTAGGTCAAC-3'); and probe (5'-ATGTCGCGCATTGGCATGGA-3').

Minor comments

1. The reviewer indicated, "The context in the Introduction 56-60 is difficult to understand in current form. These sentences should be restructured for logical flow." Originally, we discussed the utility of using saliva samples for PCR-based diagnosis of COVID-19 in the Introduction, lines 56-60. However, we acknowledge the reviewer's concern regarding the clarity of our intent. As such, we have removed the commentary on the evaluation of salivary PCR, given that the primary focus of our study is the validation of RAT for detecting SARS-CoV-2.

2. The reviewer indicated, "I cannot agree the description in the Introduction 63 "RAT for the detection of SARS-CoV-2 has received limited evaluation" based on the citation stated in 2020, since a number of studies have reported the evaluation of RAT to date." We appreciate this suggestion and agree with the reviewer's assessment. Based on the reviewer's suggestion, we have significantly changed the 2nd paragraph of the Introduction section as follows:

Page 4, line 75 to Page 5, line 90: Rapid antigen testing (RAT) using an immunochromatographic technique is now widely recognized and has been useful for the point-of-care diagnosis of COVID-19 in current clinical settings. The World Health Organization (WHO) and the Food and Drug Administration (FDA) recommend RAT kits perform with at least 80% sensitivity and have approved several kits (3). Before the emergence of the Omicron variants, the sensitivity of clinically used RAT kits with NP swab samples was approximately 70% (4,5). After the Omicron epidemic, changes in symptomatic characteristics and a reduction in severity compared to earlier strains have been documented (6,7). These alterations suggest that the virological and clinical features of COVID-19 during the Omicron period may differ from those of the pre-Omicron era. The sensitivity of RAT kits using NP swabs during the Omicron period has shown a slight decrease, though results still seem inconclusive (3,8). COVID-19 diagnosis using saliva samples offers reduced burden and risk in point-of-care settings. The

performance of RAT using saliva was examined during the pre-Omicron period; however, only a few reports are available (4). Furthermore, previous studies have shown performance discrepancies among their RAT kits using saliva (9-11).

3. The reviewer indicated, “Approval number of the IRB not specified in the text (different institution from the author's affiliation?).” We confirm that no approval number was assigned during the IRB review process, conducted at an institution separate from ours. We have updated the statement with the approval date as follows:

Page 6, lines 111 to 112: This study was approved by the institutional review board of Hara-Doi Hospital (Approval Date: January 18, 2022).

4. The reviewer indicated, “The authors described the Identification of SARS-CoV-2 variants in the Method section, whose corresponding result were not found.” We appreciate this indication. We have included the following results related to the identification of SARS-CoV-2 variants in the Results section:

Page 10, lines 190 to 192: Of the 117 SARS-CoV-2-positive samples, 105 (89.7%) were identified as Omicron variants (6 Omicron BA.1/BA.2 variants, 99 Omicron BA.4/BA.5 variants, and 12 unidentifiable strains).

5. The reviewer indicated, “About RAT used in this study, whether the use of saliva specimen is approved or not should be mentioned. How the decision was made (single or double, open or blind) should be noted.” According to the reviewer’s suggestion, we have added statements to the Introduction and Methods sections as follows:

Page 5, lines 92 to 96: The ALSONIC COVID-19 Ag immunochromatographic assay kit (Alfresa Pharma Corporation, Osaka, Japan) for NP swab samples was approved in Japan and released on March 18, 2021. It has been clinically used for point-of-care diagnosis of COVID-19 across Japan. This study formed part of a clinical trial aiming to gain additional approval for using the kit with saliva samples.

Page 7, lines 154 to 156: The determination of band positivity using the kit was performed by physicians at each participating institution, who were blinded to the PCR test results.

6. The reviewer indicated, “The statistical analysis applied for each analysis is not clear. About the analysis comparing NP and saliva viral load, paired test would fit better in this study.” We agree with this suggestion and have reanalyzed comparing NP and saliva viral loads (Figure 1) using a paired t -test. This analysis confirmed significant differences between NP and saliva viral loads (NP vs. saliva viral loads in the overall analysis, $P < 0.0001$). Consequently, we have refined our descriptions of the statistical methods as follows:

Page 8, line 175 to Page 9, line 179: Categorical variables were analyzed using the Chi-squared test. Student’s t-test was used for comparing continuous variables between two groups, while ANOVA was applied among three or more groups. A paired t-test was specifically utilized for comparing paired data between two groups. Spearman's rank-correlation test was used to assess correlations between two data sets.

7. The reviewer indicated, “Poor information in figure legends, including details about statistical analysis.” According to the reviewer’s suggestion, we have added information on statistical analysis to the Figure legend and Supplementary Figure 1 as follows:

Page 22, lines 446 to 447: Paired t-test was used to compare paired nasopharyngeal and salivary data. Supplementary Figure 1: Spearman's rank-correlation test was used to assess the correlation between nasopharyngeal and salivary data.

Responses to Reviewer 2’s comments

Major comments

1. The reviewer indicated, “The introduction does appear dated and I suspect it may have been written, at least in part, several years ago. For example, statements such as (Line 61): "RAT using an immunochromatographic technique is expected to be useful for POC diagnosis of COVID-19 in clinical settings". I would argue, this statement should be in the past tense, and that RATs have been widely used and have shown value in at-home and clinical diagnosis of COVID-19. I would also argue that the statement (Line 64) that RATs using saliva have hardly been validated is incorrect and at this point has been heavily and widely investigated across multiple technologies with multiple reviews and

metanalyses on the topic. The authors should update the introduction using more recent data and references. Because the authors focus on Omicron, additional background information on clinical presentation, viral loads, validation studies on the Omicron variant should be included (there are vast data on this topic).” We appreciate this suggestion and agree with the reviewer’s assessment. After reviewing the available literature, we acknowledge that while there is abundant data on RAT using NP swabs, the information on saliva-based RAT appears more limited, showing significant variability in validation results. Based on this and the reviewer’s suggestion, we have significantly changed the 2nd paragraph of the Introduction section as follows:

Page 4, line 75 to Page 5, line 90: Rapid antigen testing (RAT) using an immunochromatographic technique is now widely recognized and has been useful for the point-of-care diagnosis of COVID-19 in current clinical settings. The World Health Organization (WHO) and the Food and Drug Administration (FDA) recommend RAT kits perform with at least 80% sensitivity and have approved several kits (3). Before the emergence of the Omicron variants, the sensitivity of clinically used RAT kits with NP swab samples was approximately 70% (4,5). After the Omicron epidemic, changes in symptomatic characteristics and a reduction in severity compared to earlier strains have been documented (6,7). These alterations suggest that the virological and clinical features of COVID-19 during the Omicron period may differ from those of the pre-Omicron era. The sensitivity of RAT kits using NP swabs during the Omicron period has shown a slight decrease, though results still seem inconclusive (3,8). COVID-19 diagnosis using saliva samples offers reduced burden and risk in point-of-care settings. The performance of RAT using saliva was examined during the pre-Omicron period; however, only a few reports are available (4). Furthermore, previous studies have shown performance discrepancies among their RAT kits using saliva (9-11).

2. The reviewer indicated, “The only true novelty in this manuscript is the validation of rapid test for which little clinical data is available. Therefore, the authors should in their discussion, compare the performance of their rapid tests to other technologies and RATs that are more widely used and distributed - for example. does this RAT appear more, similar, or less sensitive to these other technologies? ” According to the reviewer’s suggestion, we have included additional details on the performance of the RAT used in our study, comparing it to other widely recognized technologies:

Page 14, lines 272 to 276: A recent meta-analysis evaluating the performance of RAT using NP swab samples during the Omicron period reported a sensitivity of 67.1% and a specificity of 100% (3). In addition, WHO/FDA-approved, widely utilized Abbott kits have demonstrated a sensitivity of approximately 70% with NP samples (3). Based on these reports, the RAT kit used in our study has shown excellent performance when using NP samples.

3. The reviewer indicated, “Additionally, a few additional lines in the introduction on this test should be included - was it / where is it approved, when was it released, where and how frequently is it used, etc.” According to the reviewer’s suggestion, we have added statements to the Introduction section as follows:

Page 5, lines 92 to 96: The ALSONIC COVID-19 Ag immunochromatographic assay kit (Alfresa Pharma Corporation, Osaka, Japan) for NP swab samples was approved in Japan and released on March 18, 2021. It has been clinically used for point-of-care diagnosis of COVID-19 across Japan. This study formed part of a clinical trial aiming to gain additional approval for using the kit with saliva samples.

Minor comments

1. The reviewer indicated, “The use of axillary temperature and the 37 degree cut-off for denoting fever should be justified in the methods. In the abstract and tables, when denoting fever and non-fever temperature cut-offs it should be clarified that this is through axillary measurement.” In Japan, it is common that axillary body temperature is measured and that 37 degrees is used as a cut-off for fever. Based on the reviewer’s suggestion, we have modified statements in the Methods section and have denoted axillary body temperature in the abstract and tables.

Page 7, lines 130 to 132: Axillary body temperature, routinely used in Japan, was measured at the time of sample collection. A temperature of $\geq 37^{\circ}\text{C}$ was defined as symptomatic fever.

2. The reviewer indicated, “Throughout the manuscript RT-PCR should be RT-qPCR or qRT-PCR - the RT here should denote reverse transcription not real-time as defined in the manuscript. Because of your chosen terminology some sentences are illogical like (Line 109): "Qualitative and quantitative RT-PCR detection" - as real-time and quantitative are terms that are used interchangeably. As an additional

example, you use the RT-qPCR Mix in your methods (Reverse Transcription quantitative PCR), where RT denotes Reverse” We thank for your suggestion. We have deleted the terms “qualitative”, “quantitative”, and “real-time” and have defined RT-qPCR as reverse transcription-quantitative PCR throughout the manuscript.

3. The reviewer indicated, “Line 119: What control materials were used for positive controls? Were controls used during the extraction step?” The positive control was the control RNA corresponding to the nucleocapsid region. The positive control was used in the process of viral load measurement using RT-qPCR. We have modified the statement and have cited a related reference.

Page 7, line 151 to Page 8, line 154: The viral load of each sample was calculated based on a calibration curve established using a 10-fold dilution series of the positive control RNA (nucleocapsid region) containing 1.0×10^6 copies/ μ L (14).

4. The reviewer indicated, “Line 130-135: Were test operators aware of qRT-PCR results when interpreting RATs or were results blinded? Did a single operator interpret all RAT test results?” Based on the reviewer’s inquiry, we have added statements to the Methods section as follows:

Page 8, lines 168 to 170: The determination of band positivity using the kit was performed by physicians at each participating institution, who were blinded to the PCR test results.

5. The reviewer indicated, “Where any other confounding metrics assessed that may be associated with certain symptomology (e.g. smoking with cough)?” We were unable to examine the presence or absence of confounding factors (backgrounds) that are potentially associated with symptoms, especially cough. Based on the reviewer’s indication, we have added statements to the limitation part of the Discussion section as follows:

Page 15, lines 301 to 304: We did not examine the presence or absence of backgrounds that might be associated with patient symptoms. Consequently, adjusting for potential confounding factors that could influence the reported symptoms was challenging.

6. The reviewer indicated, “Line 395: NP = Nasopharyngeal as defined in main text body.”

We have changed the explanation of the abbreviation, NP, from nasopharynx to nasopharyngeal.

Re: Spectrum00932-24R1 (Correlation of patient symptoms with SARS-CoV-2 Omicron variant viral loads in nasopharyngeal and saliva samples and their influence on the performance of rapid antigen testing)

Dear Dr. Yong Chong:

Thank you for the privilege of reviewing your work. Below you will find my comments, instructions from the Spectrum editorial office, and the reviewer comments.

Firstly, some questions have been raised regarding the suitable ethical approval for the conducted work. Please do be sure to include an appropriate ethics statement and attend to reviewer comments regarding clarifications of methodology in your response.

Revision Guidelines

Sincerely,
Sophia Georghiou
Editor
Microbiology Spectrum

Reviewer #1 (Comments for the Author):

I thank the authors for revising the manuscript. I believe the authors responded to the comments appropriately and agree the revised version of the manuscript and support the importance of this study.

Reviewer #2 (Comments for the Author):

It was with interest that I reviewed the manuscript entitled "Correlation of patient symptoms with SARS-CoV-2 Omicron variant viral loads in nasopharyngeal and saliva samples and their influence on the performance of rapid antigen testing" resubmitted to Microbiology Spectrum. This manuscript presents a study on the correlation between SARS-CoV-2 viral loads in nasopharyngeal (NP) and saliva samples, patient symptoms, and the performance of rapid antigen testing (RAT) during the Omicron variant period in Japan. The study is relevant given the ongoing need for effective point-of-care diagnostics, especially in the context of evolving SARS-CoV-2 variants. The study was well-designed and it appears the appropriate statistical tests were used in analyses. The manuscript is well-written and easy to follow.

The authors have taken time to address many of the comments raised by previous reviewers. I do have several additional comments, that may further improve clarity and help provide context for this work.

Major Comments:

1. Conflict of Interest Statement: The manuscript acknowledges funding by Alfresa Pharma Co., Ltd. in the Acknowledgements section, but it lacks a clear conflict of interest statement. Given that the research was funded by the test manufacturer, a specific conflict of interest statement should be included.
2. Rationale for Including Only the First Four Days of Symptom Onset: The manuscript does not clearly explain the rationale for including only the first four days of symptom onset in the study. It is important to clarify whether this decision is based on regulatory instructions, approval guidelines, or a specific scientific rationale. This explanation should be included in the Methods section to provide context for this decision.
3. It is also critical that the manuscript clearly states, both in the text and in the tables, that the reported sensitivity values reflect tests conducted within the first four days post-symptom onset. This clarification will prevent any potential misinterpretation of the data by readers.
4. Ethics Statement: The manuscript should include an ethics statement.
5. Consider elaborating further on the rationale for using salivary RAT in clinical settings, particularly in symptomatic versus asymptomatic populations. The study fails to frame the importance/utility of saliva-based RATs versus nasal swabs or other methods.

Minor Comments:

1. Table 2 currently lacks n values for positive and negative cases. Including these values is crucial for readers to fully understand the context of the sensitivity and specificity results. This information should be added to Table 2.
2. Line 138 of the manuscript refers to "RT-PCR" instead of "RT-qPCR."
3. The manuscript does not specify the fluorophore (FAM?) used in the RT-qPCR assay on Line 149.
4. On Line 154, please indicate the source of the oligos used in the RT-qPCR assay.

Responses to Reviewer 2's second comments

Major comments

1. The reviewer indicated, "Conflict of Interest Statement: The manuscript acknowledges funding by Alfresa Pharma Co., Ltd. in the Acknowledgements section. Given that the research was funded by the test manufacturer, a specific conflict of interest statement should be included." We thank you for the suggestion. Accordingly, we have independently described the conflict of interest in the Acknowledgements section.

2, 3. The reviewer indicated, "Rationale for Including Only the First Four Days of Symptom Onset: The manuscript does not clearly explain the rationale for including only the first four days of symptom onset in the study. It is important to clarify whether this decision is based on regulatory instructions, approval guidelines, or a specific scientific rationale. This explanation should be included in the Methods section to provide context for this decision." and "It is also critical that the manuscript clearly states, both in the text and in the tables, that the reported sensitivity values reflect tests conducted within the first four days post-symptom onset. This clarification will prevent any potential misinterpretation of the data by readers." We have described the inclusion of the first three days in the Abstract and Results section of the original manuscript as "A total of 153 patients tested within three days of symptom onset were included." and "39 patients tested ≥ 4 days after the onset of symptoms were excluded.", respectively. Of the 39 patients, 19 were SARS-CoV-2-positive, and the number of positive cases by date after the four days of symptom onset was < 10 . Therefore, we excluded cases tested ≥ 4 days after the onset of symptoms. We have explained the exclusion process in the Results section. We have defined the number of days from symptom onset in the Methods section, modified statements regarding the exclusion process in the Results section, and added the inclusion criteria to the footnotes of the tables.

Page 7, lines 131 to 132 : The day of symptom onset was defined as day 1 after symptom onset.

Page 10, lines 190 to 192: 39 patients tested ≥ 4 days (day 4) after the onset of symptoms were excluded because of the small number of SARS-CoV-2-positive cases (< 10 cases per day).

4. The reviewer indicated, "Ethics Statement: The manuscript should include an ethics statement."

Based on the reviewer's indication, we have transferred the ethics statement in the participants part of

the Methods section to the “Ethics Approval” section following the main text.

5. The reviewer indicated, “Consider elaborating further on the rationale for using salivary RAT in clinical settings, particularly in symptomatic versus asymptomatic populations. The study fails to frame the importance/utility of saliva-based RATs versus nasal swabs or other methods.” Based on the reviewer’s suggestion, we have modified the commentary on the importance/utility of using saliva as RATs in the Introduction section. Additionally, the results of this study reflect the utility of salivary RAT in the presence or absence of symptoms. The sensitivity of salivary RAT in symptomatic patients reached 80%. We have further discussed the utility of salivary RAT in symptomatic patients in the Discussion section.

Page 4, line 86 to Page 5 line 91: There are several advantages to using saliva for sample collection rather than NP swabs. NP swabs are needed to be collected by trained healthcare personnel wearing suitable protective equipment to prevent secondary infections. The risk of infection may be higher, particularly in patients with symptoms. In contrast, collecting saliva samples could contribute to reducing the burden and risk to healthcare personnel because self-collection is possible.

Page 15, lines 302 to 305: The risk of secondary infections during sample collection is potentially higher in symptomatic patients than in asymptomatic patients. The diagnosis of COVID-19 using salivary RAT may be useful, particularly in patients with symptoms.

Minor comments

1. The reviewer indicated, “Table 2 currently lacks n values for positive and negative cases. Including these values is crucial for readers to fully understand the context of the sensitivity and specificity results. This information should be added to Table 2.” According to the reviewer’s suggestion, we have added n values for the positive and negative cases in Table 2.

2. The reviewer indicated, “Line 138 of the manuscript refers to "RT-PCR" instead of "RT-qPCR."” We have changed RT-PCR to RT-qPCR in the Methods section.

3. The reviewer indicated, “The manuscript does not specify the fluorophore (FAM?) used in the RT-qPCR assay on Line 149.” According to the reviewer’s indication, we have modified the base sequence of the probe as follows:

Page 7, lines 152 to 153: probe (5'-FAM-ATGTCGCGCATTGGCATGGA-BHQ-3')

4. The reviewer indicated, “On Line 154, please indicate the source of the oligos used in the RT-qPCR assay.” We have described the source of the positive control RNA as follows:

Page 8, lines 156 to 159: The viral load of each sample was calculated based on a calibration curve established using a 10-fold dilution series of the positive control RNA (nucleocapsid region) containing 1.0×10^6 copies/ μL with the Positive Control RNA Mix (2019-nCoV) (Takara Bio Inc., Kusatsu, Japan) (14).

Re: Spectrum00932-24R2 (Correlation of patient symptoms with SARS-CoV-2 Omicron variant viral loads in nasopharyngeal and saliva samples and their influence on the performance of rapid antigen testing)

Dear Dr. Yong Chong:

Your manuscript has been accepted, and I am forwarding it to the ASM production staff for publication. Your paper will first be checked to make sure all elements meet the technical requirements. ASM staff will contact you if anything needs to be revised before copyediting and production can begin. Otherwise, you will be notified when your proofs are ready to be viewed.

Sincerely,
Sophia Georghiou
Editor
Microbiology Spectrum

Reviewer #2 (Comments for the Author):

The authors have successfully addressed the comments raised by previous reviewers, and I have no additional recommendations or concerns.